# Inundation mapping based on reach-scale effective geometry

Cédric Rebolho[1], Vazken Andréassian[1], and Nicolas Le Moine[2]

[1]Irstea, UR HYCAR, 1 Rue Pierre-Gilles de Gennes, 92160 Antony, France
[2]Sorbonne Universités, UPMC Univ Paris 06, CNRS, EPHE, UMR 7619 Metis, 4 Place Jussieu, 75005 Paris, France

*Correspondence to:* Cédric Rebolho (cedric.rebolho@irstea.fr)

**Abstract.** The production of spatially accurate representations of potential inundation is often limited by the lack of available data as well as model complexity. We present in this paper a new approach for rapid inundation mapping, MHYST, which is well adapted for data-scarce areas; it combines hydraulic geometry concepts for channels and DEM data for floodplains. Its originality lies in the fact that it does not work at the cross section scale but computes effective geometrical properties to describe the reach scale. Combining reach-scale geometrical properties with 1-D steady-state flow equations, MHYST computes a topographically coherent relation between the "Height Above Nearest Drainage" and streamflow. This relation can then be used on a past or future event to produce inundation maps. The MHYST approach is tested here on an extreme flood event that occurred in France in May-June 2016. The results indicate that it has a tendency to slightly underestimate inundation extents, although efficiency criteria values are clearly encouraging. The spatial distribution of model performance is discussed and it shows that the model can perform very well on most reaches, but has difficulties modelling the more complex, urbanised reaches. MHYST should not be seen as a rival to detailed inundation studies, but as a first approximation able to rapidly provide inundation maps in data-scarce areas.

## 1 Introduction

Floods are a recurring phenomenon in France: in September 2014, intense rainfall affected the south of the country, leading to several deaths and about 0.6 billion euros worth of damage. The following year, in October, about 20 people died in the southeast due to massive flooding, which caused a loss of half a billion euros. Then, in June 2016, large-scale flooding occurred over the Seine and Loire catchments, mainly affecting their tributaries and resulting in four deaths at a cost of 1.4 billion euros. These are only examples which underline the value of flood inundation mapping to anticipate the impact of such events. Public authorities and insurance companies are showing a growing interest in the field of rapid inundation modelling, and for the development of simple methods, that would work for any river with easily available data.

Flood hazard assessment usually combines rainfall observations or simulations, a hydrological model, streamflow simulations or observations, and an inundation model in order to generate inundation extents, height maps and sometimes other information (e.g. velocities). Traditionally, flood inundation models are derived from the Shallow Water Equations (SWE) in one or two dimensions (the so-called hydraulic models), with various simplifications that have proved to give satisfying results. For instance, the Regional Flood Model (RFM), probably one of the most comprehensive approaches published so far, is made of four parts (Falter et al., 2014): a daily distributed rainfall-runoff model, a 1D hydraulic model for channel routing, a 2D

hydraulic model for floodplain mapping and a flood loss estimation model. Its application on the Mulde catchment in Germany (Falter et al., 2015) showed mixed results concerning inundation extents, correctly predicting only 50% of the flooded area for the August 2002 event. This underestimation was explained by dike breaches that were not accounted for within the model. Lack of observed data did not allow validation on other events.

Not all hydraulic models need to have this degree of complexity. It is indeed possible to neglect specific parts of the SWE depending on the situation. Usually, 2D models use the complete Saint-Venant equations while 1D models often disregard one or several terms, leading, for instance, to the diffusive wave or kinematic wave approximations (e.g. Moussa and Cheviron, 2015). Some methods choose to couple 1D and 2D models, the former for streamflow routing and the latter for overbank flow (Morales-Hernández et al., 2016). Despite the accuracy of such models, studies often try to further simplify them because of

the large computing time to simulate small areas and the lack of precise data required to run these models.

LISFLOOD-FP (Bates and De Roo, 2000), a hydraulic model developed to simulate floodplain inundation, was used in several studies (Horritt and Bates, 2001; Hunter et al., 2005; Biancamaria et al., 2009). The model offers different possibilities: using 2D equations or 1D equations decoupled on a 2D grid with kinematic, diffusive or inertial approximations (Bates et al., 2010). Horritt and Bates (2002) published a comparison between different models with gradually increasing complexities (1D,

1D on 2D grid and 2D) and, surprisingly, showed that the 1D model had a better ability to reproduce the two events that were used in validation. The subsequent analysis concluded that the reach studied was relatively narrow and could easily be modelled using simple methods, and the authors argued that the other models would be more appropriate for more complex reaches.

However, these examples concern relatively small and well-instrumented reaches and assessing flood hazard at a larger scale

may require different approaches. Alfieri et al. (2014) applied LISFLOOD-ACC, an inertial version of LISFLOOD-FP with decoupled 1D equations on a 100-m resolution grid over Europe in order to map flood hazard for a 100-years return period, assuming a constant return period along the reaches. Broadly speaking, the model splits rivers into small reaches, to apply the hydraulic models independently and to merge simulated maps together, but only for rivers with a catchment larger than 500 km$^2$. The model was then validated against regional and national hazard maps for six catchments in Germany and the

United Kingdom and showed a general over-prediction. Another variation of LISFLOOD-FP for large-scale flood inundation modelling was introduced by Neal et al. (2012), including a new subgrid representation of channel networks for improved model accuracy (Neal et al., 2012; Schumann et al., 2013).

Le Bihan et al. (2017) developed an approach aimed at the forecasting context, in order to cope with excessive computing times. The solution chosen was to run a simple 1D hydraulic model during a "pre-analysis phase" and create a catalogue of

inundation extents corresponding to various return periods. These maps are then used, in a forecasting context, to give an estimate of the level of flooding, depending on the forecast discharge.

The lack of precise data (especially for channel cross sections) and the computing time required by numerical methods for solving the SWE motivated the development of potentially alternative methods, mostly based on DEM analysis. For instance, the Rapid Flood Spreading Method (RFSM, Gouldby et al., 2008) chose to divide floodplains into impact zones of different

elevations in order to explore the effects of dike breaches using a spilling algorithm based on water depth. Other methods

derive inundation maps from topographic information only: one can cite EXZECO (Pons et al., 2010), which introduces elevation noise in the DEM in order to create a single map of "maximum flow accumulation" that can be seen as a potential inundation area, and HAND (Height Above Nearest Drainage), a descriptor originally used for terrain classification (Rennó et al., 2008; Nobre et al., 2011), which has recently been adapted to static flood inundation mapping (Nobre et al., 2016) and is increasingly used to produce flood maps (e.g. Afshari et al., 2018; Speckhann et al., 2018; McGrath et al., 2018). HAND calculates the difference between river cells' elevation and that of the connected floodplain cells, thus giving relative height information which can be compared to observed flood depths and the corresponding inundation extent.

MHYST, the method presented in this paper, is a simplified approach developed with the aim of rapidly producing inundation maps in data-scarce areas. It combines (i) concepts of hydraulic geometry to characterise channel geometry and (ii) DEM-derived relative elevations to characterise the floodplain; it does not work at the cross section scale but computes effective geometrical properties representative of the reach scale. Combining reach-scale geometrical properties with simplified steady-state hydraulic laws allows one to rapidly generate flood inundation maps while ensuring reach-scale coherence. After describing the method and the calibration dataset, MHYST is compared against the inundation extent observed for the major event that occurred in May-June 2016 in France. The last section discusses the spatial distribution of performance and the impact of uncertainties on the results obtained.

## 2   MHYST : a simplified steady-state hydraulic approach

The MHYST model stands for *Modélisation HYdraulique simplifiée en écoulement STationnaire*, i.e., *Simplified Steady-state Hydraulic Modelling*. It is a flood inundation model which aims to map inundation extents at the reach scale. Where classic hydraulic models use cross sections, this method is based on an effective geometry representative of each river reach. Since no detailed geometric data were available to describe the shape and roughness of the channel river bed for this study, a subgrid representation of the channel was derived from hydraulic geometry relationships linking drainage area with bankfull width and height (Leopold and Maddock, 1953). When discharge exceeds bankfull capacity, the model computes a reach-scale relation between streamflow and the "Height Above Nearest Drainage" (HAND) defined by Nobre et al. (2016). This relation can finally be used to assess which height corresponds to the given streamflow, and thus to derive the corresponding inundation map.

### 2.1   Processing of DEM: from elevations to Height Above Nearest Drainage

The initial step consists in processing the Digital Elevation Model (DEM) in order (i) to obtain a flowing Drainage Direction map, (ii) to identify the subcatchments (corresponding to the river reaches), and (iii) to compute the Height Above Nearest Drainage (HAND) in each subcatchment. This initial processing is the basis of the floodplain analysis in MHYST. To compute the Drainage Direction map, we used the D8 method from the Flow Direction function provided by ArcGIS 10.3. It computes the drainage direction by calculating the steepest slope from the eight possible directions for a given cell.

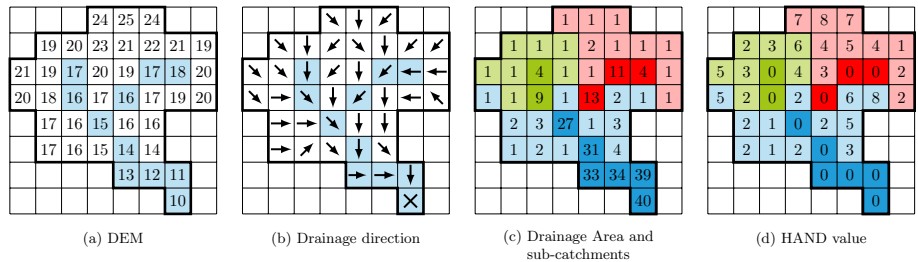

| (a) DEM | (b) Drainage direction | (c) Drainage Area and sub-catchments | (d) HAND value |

**Figure 1.** Processing of DEM and calculation of the HAND value for a hypothetical catchment.

Figure 1 shows the procedure used to compute HAND values: for a given floodplain cell, it is the difference between its elevation and that of the closest river cell in terms of drainage direction. For instance, the cell of elevation 25 at the top of the figure is linked to (i.e. flows towards) the most upstream red river cell which has an elevation of 18: thus, its HAND value is $25 - 18 = 7$. This relative height has been used as a proxy for inundation height by various studies (Nobre et al., 2016; Afshari et al., 2018). To derive an inundation map from HAND values, we must define a threshold height $H_T$: the flooded area corresponds to all the cells whose HAND value is strictly lower than $H_T$ (Jafarzadegan and Merwade, 2017).

## 2.2 Model description

MHYST is mostly based on a DEM and its derivatives (drainage map and drainage areas) and on the hydraulic equations describing a steady uniform flow at the reach scale. This means that for a given time-step (day in this case), at a given reach, we make the approximation that the flow is constant over time and space (this is obviously a strong simplification that we will discuss later). Table 1 sums up the variables used in the following equations as well as their respective units and interpretations. Table 2 describes the two free parameters of the model. The following equations show the path to build a reach-scale relation between $H_T$ and the streamflow $Q$ by calculating, with hydraulic formulas, the discharge value corresponding to a given $H_T$. Once this relation is known, the model can easily simulate a hydrological event by inverting the relation, and by searching for which $H_T$ corresponds to the given $Q$ (Fig. 2).

Other variables can be directly calculated from the DEM (Fig. 4): for a given threshold height $H_T$ at a reach of length $L$ ($L$ is a fixed parameter of the model), $V(H_T)$ is the sum of volumes above all flooded pixels and $S(H_T)$ is the area occupied by the flooded cells. $A(H_T)$, in Eq. 1, is the average cross section area of the flooded reach and it depends on $V(H_T)$ and on the bankfull cross section area of the channel ($A_b = h_b \cdot W_b$, Fig. 3). This variable can also be defined as the sum of the channel cross section area ($A_{ch} = A_b + H_T \cdot W_b$) and the floodplain cross section area ($A_{fp} = A(H_T) - A_{ch}$). $B(H_T)$, in Eq. 2, is the average surface width of the flooded reach, defined similarly from $S(H_T)$ and $L$.

$$A(H_T) = \frac{1}{L} \cdot V(H_T) + A_b = A_{ch} + A_{fp} \tag{1}$$

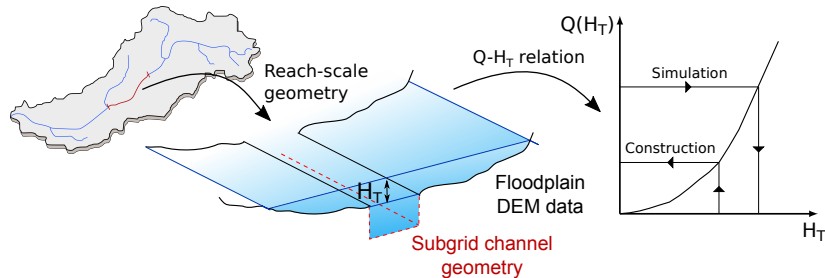

**Figure 2.** Representation of the model structure: the reach-scale geometry is derived from hydraulic geometry relationships and DEM data and is then used to compute a relation between the threshold height $H_T$ and the discharge $Q$. $L$ is a fixed characteristic of the reach.

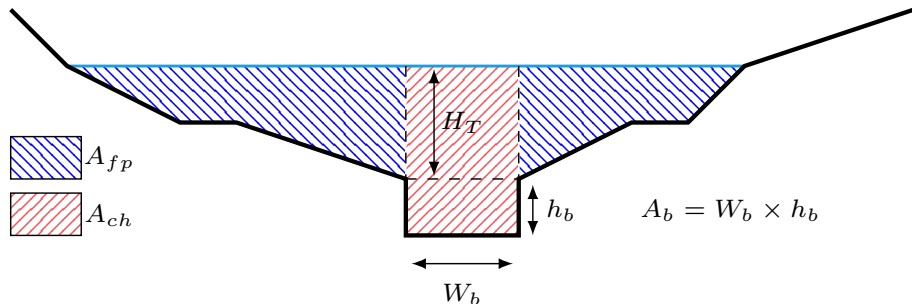

**Figure 3.** Typical cross-section segmentation, with the cross-section area of the channel ($A_{ch}$), that of the floodplains ($A_{fp}$) and the bankfull cross-section area ($A_b$) which is calculated from the average bankfull height ($h_b$) and width ($W_b$) computed from downstream hydraulic geometry relationships.

$$B(H_T) = \frac{1}{L} \cdot S(H_T) \tag{2}$$

The only unknown variables in these equations are subgrid parameters $h_b$ (bankfull water level) and $W_b$ (bankfull width), i.e. the bankfull geometry, which cannot be obtained from usual DEMs and are only available from detailed surveys for a small number of rivers. Indeed, *in situ* bathymetric data are quite scarce and red lasers cannot penetrate the water surface more than a few centimeters, which means that the real elevation of the river bed is mostly not correctly represented in the DEMs. This is why we chose to use downstream hydraulic geometry equations to estimate these geometric parameters, assuming a rectangular channel, the size of which depends on the upstream drainage area (Eq. 3 and 4). To assess the coefficients $\alpha$ and $\beta$, we used satellite images from the French platform Géoportail in order to link observed bankfull widths and drainage areas. The values found for the Loing catchment are: $\alpha = 0.053$ and $\beta = 0.822$. The other coefficients, $\delta$ and $\omega$, were taken from a study by Blackburn-Lynch et al. (2017), which attempted to regionalise these parameters in the U.S. We used the general

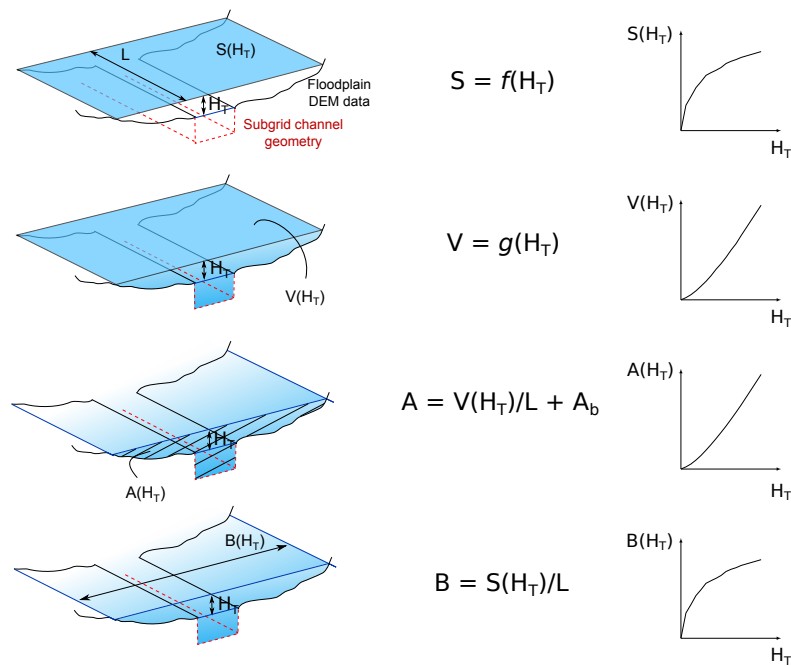

**Figure 4.** Representation of the reach-scale geometry derived from HAND and the DEM. $A(H_T)$ and $B(H_T)$ are derived from $V(H_T)$ and $S(H_T)$ respectively (Eq. 1 and 2).

values found for the whole set of catchments: $\delta = 0.27$ and $\omega = 0.21$. Although these values probably add uncertainties in the model, they are an accessible way to assess bankfull channel geometry and could still be improved by local bankfull studies when available.

$$W_b = \alpha \cdot A_D^{\beta} \tag{3}$$

$$h_b = \delta \cdot A_D^{\omega} \tag{4}$$

The fundamental equations of the MHYST model come from an experimental study by Nicollet and Uan (1979) which defines the DEBORD formulation as in Eqs. 5 to 7. Building on the Manning-Strickler formula, these authors proposed an empirical parameterisation of turbulent momentum exchange between the channel and the floodplain. This formulation expresses the conveyance capacity depending on channel-related and floodplain-related variables. The coefficient C takes into account

the interaction of flows between the fast-flowing channel and the slow-flowing floodplain, and the corresponding head losses.

$$De = K_{ch} \cdot C \cdot A_{ch} \cdot R_{ch}^{2/3} + K_{fp} \cdot \sqrt{A_{fp}^2 + A_{ch} \cdot A_{fp} \cdot (1 - C^2)} \cdot R_{fp}^{2/3} \tag{5}$$

$$C = \begin{cases} C_0 = 0.9 \cdot \left( \dfrac{K_{fp}}{K_{ch}} \right)^{1/6} & \text{if } r = \dfrac{R_{fp}}{R_{ch}} > 0.3 \\ \dfrac{1 - C_0}{2} \cdot \cos\left( \dfrac{\pi \cdot r}{0.3} \right) + \dfrac{1 + C_0}{2} & \text{if } 0 \leq r \leq 0.3 \end{cases} \tag{6}$$

$$Q = De \cdot \sqrt{I_f} \tag{7}$$

The streamflow $Q$ is finally defined from the conveyance capacity and the channel slope, since we hypothesise a uniform flow. $R_{ch}$ and $R_{fp}$ can easily be calculated from the assumed reach geometry (Eq. 8 and 9), which only leaves the Strickler coefficients as unknown variables.

$$R_{ch} = \frac{A_{ch}}{W_b + 2 \cdot h_b} \tag{8}$$

$$R_{fp} = \frac{A_{fp}}{B(H_T) - W_b} \tag{9}$$

The two Strickler coefficients add two degrees of freedom, and $K_{ch}$ is additionally used to calculate the bankfull flow from the Manning-Strickler formula (Eq. 10).

$$Q_b = K_{ch} \cdot \left( \frac{W_b \cdot h_b}{L_b + 2 \cdot h_b} \right)^{2/3} \cdot \sqrt{I_f} \cdot h_b \cdot W_b \tag{10}$$

Here, we sum up the procedure, which operates at the reach scale:

1. For a given threshold height $H_T$, we use the DEBORD formulation to calculate the corresponding discharge $Q$.

2. By repeating the operation for all possible $H_T$, we obtain a reach-specific table matching values of $H_T$ and $Q$.

3. When working on an event where only $Q$ is known, when it is greater than $Q_b$ (which means that the river overflowed), the model looks for the corresponding $H_T$ value in the table, by calculating a linear interpolation between two values if necessary, and then assigns to each cell in the subcatchment a flooded height $h_{flood} = max(0; H_T - HAND_{\text{cell}})$.

Although this method and that of Zheng et al. (2018) were developed independently, they share a lot of similarities, both using HAND to derive a reach-scale geometry which is used as input for a simplified hydraulic model. However, in addition to HAND, MHYST uses downstream hydraulic geometry relationships to evaluate a subgrid representation of the channel geometry. The hydraulic model is also different: Zheng et al. (2018) use the Manning-Strickler formula, while MHYST computes streamflow values from the DEBORD formulation.

**Table 1.** Names, units and interpretations of the variables used in the geometric and hydraulic equations of the MHYST model.

| Variable | Unit | Interpretation |
|:---:|:---:|:---:|
| $H_T$ | m | Threshold height |
| $V(H_T)$ | m$^3$ | Volume created by a height $H_T$ over a reach |
| $S(H_T)$ | m$^2$ | Flooded area created by a height $H_T$ over a reach |
| $A(H_T)$ | m$^2$ | Average cross section area created by a height $H_T$ over a reach |
| $B(H_T)$ | m | Average surface width created by a height $H_T$ over a reach |
| $Q(H_T)$ | m$^3 \cdot$s$^{-1}$ | Mean discharge created by a height $H_T$ over a reach |
| $D_e$ | m$^3 \cdot$s$^{-1}$ | Conveyance capacity |
| $A_{ch}$ | m$^2$ | Cross section area of the channel |
| $A_{fp}$ | m$^2$ | Cross section area of the floodplain |
| $R_{ch}$ | m | Hydraulic radius of the channel |
| $R_{fp}$ | m | Hydraulic radius of the floodplain |
| $I_f$ | m$\cdot$m$^{-1}$ | Slope of the channel |
| $h_b$ | m | Bankfull water level of the channel |
| $W_b$ | m | Bankfull width of the channel |
| $A_b$ | m$^2$ | Bankfull cross section area of the channel |
| $Q_b$ | m$^3 \cdot$s$^{-1}$ | Bankfull discharge of the channel |
| $A_D$ | m$^2$ | Drainage area upstream a given cell |
| $L$ | m | Target length of a reach (fixed) |

**Table 2.** Names, units and interpretations of the free parameters of MHYST's structure.

| Parameter | Unit | Interpretation |
|:---:|:---:|:---:|
| $K_{ch}$ | m$^{1/3} \cdot$s$^{-1}$ | Strickler roughness coefficient for the channel |
| $K_{fp}$ | m$^{1/3} \cdot$s$^{-1}$ | Strickler roughness coefficient for the floodplain |

## 2.3 Boundary conditions

MHYST can work with either simulated or observed flows. In this paper, observed data from 12 measurement stations of the French HYDRO database (Leleu et al., 2014) were used to create an observed distributed streamflow map by interpolating flows based on drainage area (Eq. 11) for river pixels between outlets:

$$Q = Q_{up} + \frac{A_D - A_{D,up}}{A_{D,down} - A_{D,up}} \times (Q_{down} - Q_{up}) \tag{11}$$

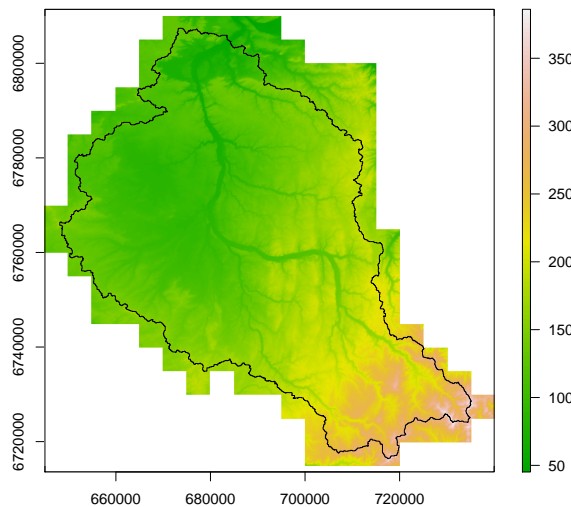

**Figure 5.** 5-m depressionless DEM used in this study. Elevations go from $45$ to $390$ m. Corrections have been applied so that each pixel flows to the sea.

where $Q$ and $A_D$ are the streamflow and drainage area of any river cell between two outlets, $Q_{up}$, $Q_{down}$, $A_{D,up}$ and $A_{D,down}$ are the direct upstream and downstream outlet discharges and drainage areas. This way, streamflow is coherently interpolated over the network, and then averaged at the reach scale.

## 3  Material

### 3.1  Generic data

In this study, we used a $5-\mathrm{m}$ resolution DEM with a vertical resolution of $0.01$m covering the Loing catchment (Fig. 5) from IGN (the French national institute for geographic information), which was filled and corrected to avoid depressions and to allow a strict coherence of flow directions, meaning that every pixel flows to the sea. Drainage directions and areas were derived from this DEM and used as model inputs along with elevations. The adaptations and modifications of the DEM were conducted using ESRI ArcGIS 10.3.

Daily observed discharges were obtained from the French HYDRO database (Leleu et al., 2014) and the stations were used to delineate the hydrological network over the catchment. Calibration data for the Loing catchment were obtained from the activation EMSN028 of the Copernicus Emergency Management Service (©2016 European Union). The original Copernicus study covered a small part of the River Seine and half of the Loing catchment (Fig. 6). However, since the study area and the defined river network were smaller, we cropped the inundation extent to match the study area (Fig. 6). These calibration data

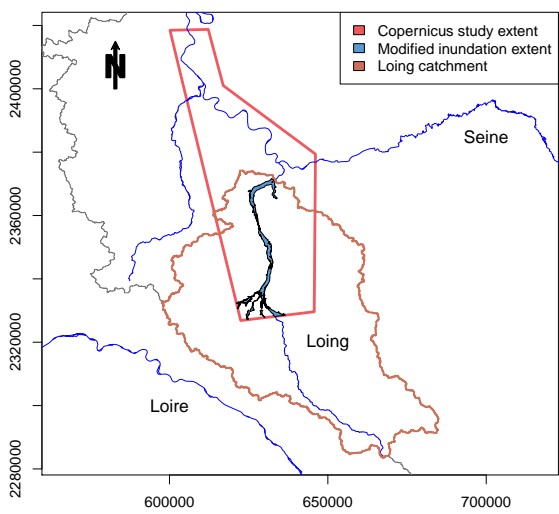

**Figure 6.** Maximum flood extent for the May-June 2016 event over the Loing catchment produced by the Copernicus Emergency Management Service.

are post-processed observed data, meaning that the original maps came from satellite observations but they were then modified to build a more homogeneous inundation extent, i.e. nearby areas whose elevations were below the observed flood level were added to the inundation extent and merged with all the others. The maximum flood extent was then validated by the European Service against reported flood damage and hydrological measurements (SERTIT, 2016).

## 3.2 Event of May-June 2016

Following an extremely wet month of May (namely the wettest on record for many stations), a heavy rainfall event started on May 30, 2016 over the center of France, affecting the Upper and Middle Seine basin and the Middle Loire basin. This episode lasted until June 6 and, combined with highly saturated soils due to a series of preceding minor events, led to major flood inundations. Over this period, overall precipitation reached $180\,\mathrm{mm}$ in Paris and Orléans, while in some tributaries, such as the River Loing, peak flows largely exceeded those of the record 1910 flood event (Fig. 7). The flood resulted in four deaths, 24 people injured and 1.4 billion euros worth of damage. A total of 1 148 cities were declared in a state of natural disaster and insurance companies received about 182 000 claims (CCR, 2016).

Since calibration data were available for June 2016 event, we chose to use our model to simulate this episode and compare the results with observations. We conducted this study over the River Loing, tributary to the River Seine, with a catchment covering $3\,900\,\mathrm{km}^2$, a mean elevation of $148\,\mathrm{m}$ and a mean slope of $0.03\,\mathrm{m}\cdot\mathrm{m}^{-1}$. This catchment was heavily impacted by

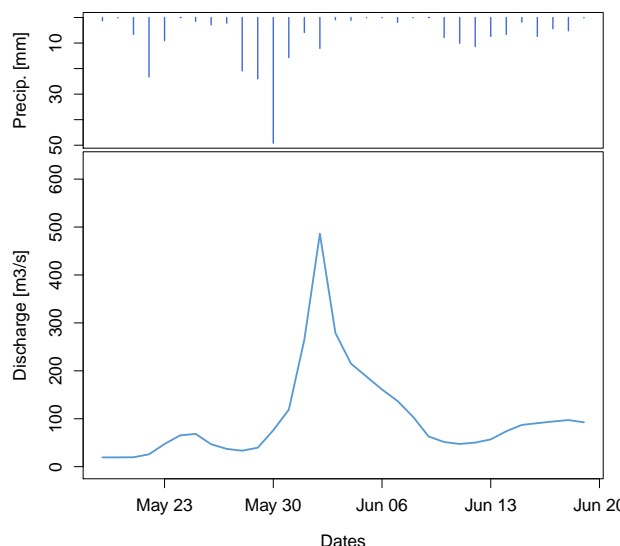

**Figure 7.** Daily hydrograph of the River Loing at Épisy ($3\,900\,\mathrm{km}^2$) during the event of June 2016. Overall precipitation reached $130\,\mathrm{mm}$. The peak discharge was the largest ever observed on the catchment and reached about $500\,\mathrm{m}^3 \cdot \mathrm{s}^{-1}$.

the flood event and concentrates a significant proportion of the inundated area, making it a suitable area to carry out the study. Streamflow data were interpolated from measurements, so no hydrological model is involved in this paper.

## 4   Results

### 4.1   Calibration procedure

To assess the model's performance, we used several criteria based on the contingency table in Fig. 8. These scores are presented in detail by Jolliffe and Stephenson (2003) and are defined as a ratio between members of the table where $n_1$ is the number of hits, i.e. the number of flooded cells correctly forecast, $n_4$ is the number of pixels correctly forecast as dry, $n_2$ is the number of false alarms and $n_3$ the number of observed flooded cells missed by the model. Table 3 summarises the formulas and the interpretations of each score used in this study.

The $POD$ (Probability of detection), which is also called $Correct$ (Alfieri et al., 2014) or $M1$ (Teng et al., 2015), calculates the percentage of observed inundated pixels intersected by the simulation map. Its main drawback is that it does not take into account the false alarms and thus it can give good results for a clearly overstimating inundation extent. On the contrary, the $FAR$ (False alarm ratio) or $M2$ (Teng et al., 2015) computes the proportion of cells wrongly flooded by the model. But similarly, if the model does not flood anything, the $FAR$ can reach its optimal value. The $CSI$, also known as $Fit$, $F$ index or

$FAI$ (Alfieri et al., 2014; Bates and De Roo, 2000; Falter et al., 2015), is a criterion which tries to give an overall performance

| | | Observed | |
|---|---|---|---|
| | | Flood | Dry |
| Model | Flood | Hits ($n_1$) | False alarms ($n_2$) |
| | Dry | Misses ($n_3$) | Correct negative ($n_4$) |

**Figure 8.** Contingency table gathering the different scenarios encountered during calibration (the numbers refer to pixels).

of the simulation by calculating the percentage of correctly flooded cells above the total number of flooded cells (observed and simulated). In this way, the score is penalised by the over- and under-estimation. However, this criterion does not specify if the model is over- or under-estimating the observed extent. This is why we also looked at the $BIAS$, which computes the ratio between the number of simulated and observed flooded cells. If it is above 1, the model over-estimates, and if it is below 1, it under-estimates. However, a value of 1 does not equal a perfect simulation since there may be a balance between the misses and the false alarms.

These ratios are particularly reliable if they are used to compare simulations and exhaustive observations. This is almost the case with Copernicus calibration data which represent a "maximum flood extent". However, MHYST outputs are dated, which is not the case for the observed map. This is why all daily simulated inundation extents were merged into one maximum simulated extent, meaning that we did not try to validate the temporal dynamic of the flood, but only aimed to assess its largest area. Thus, the preceding scores will only evaluate MHYST's ability to reproduce the maximum flood extent.

## 4.2 Parameterisation

MHYST has two free parameters (Table 2): $K_{ch}$ (the Strickler roughness coefficient for the channel) and $K_{fp}$ (the Strickler roughness coefficient for the floodplains). Preliminary studies showed that, for the Loing catchment, a length of 1 000 m was a good trade-off between accuracy and computation time; consequently L was fixed at 1 000 m in the rest of this study. $K_{ch}$ and $K_{fp}$ values were tested in the range $[\,0.1\,;\,30\,]$ in order to explore a wide range of possibilities (121 combinations were tested).

To help make a decision on the optimal parametrisation of the model, we used the following graphs, on which each ($K_{ch}$, $K_{fp}$) couple is characterised by one overall value:

– Two contour plots (Fig. 9) showing the impact of $K_{ch}$ and $K_{fp}$ for the two main scores ($BIAS$ and $CSI$);

– A Pareto plot (Fig. 10.a) showing the role played by the $K_{fp}$ parameter in balancing the $POD$ and the $FAR$;

**Table 3.** Table of forecast scores used to assess the performance of a flood simulation. All criteria are based on the contingency table (Fig. 8) and reflect one characteristic of the model. Taken together, they provide a comprehensive analysis of the model's behaviour.

| Score | Ratio | Range | Perfect score | Characteristics |
|---|---|---|---|---|
| Bias ($BIAS$) | $\dfrac{n_1 + n_2}{n_1 + n_3}$ | $[0, +\infty[$ | 1 | Measures the over-estimation ($BIAS > 1$) and under-estimation ($BIAS < 1$) of the model. |
| False alarm ratio ($FAR$) | $\dfrac{n_2}{n_1 + n_2}$ | $[0, 1]$ | 0 | Fraction of flooded pixels that were actually observed dry. Ignores misses. |
| Probability of detection ($POD$) | $\dfrac{n_1}{n_1 + n_3}$ | $[0, 1]$ | 1 | Proportion of flooded cells intersected by the model. Ignores false alarms. |
| Critical success index ($CSI$) | $\dfrac{n_1}{n_1 + n_2 + n_3}$ | $[0, 1]$ | 1 | Counts the number of correct flooded cells, while penalising over-estimation (false alarms) and under-estimation (misses). |

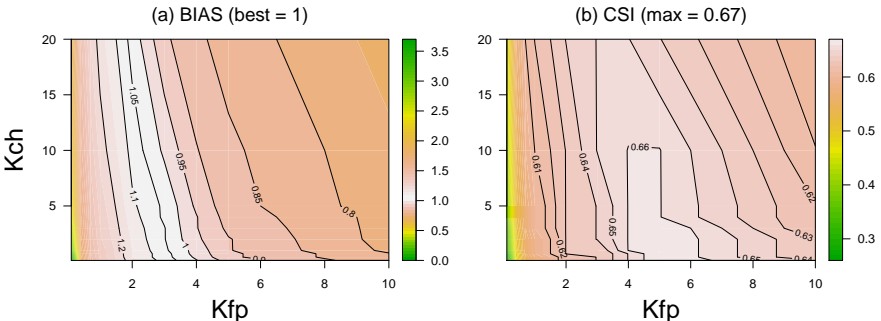

**Figure 9.** Forecast scores obtained by the model on the River Loing versus Copernicus data for all the parameter values tested, (a) $BIAS$ contour lines and (b) $CSI$ contour lines for various values of $K_{ch}$ and $K_{fp}$.

- A Pareto plot (Fig. 10.b) showing that the $CSI$ identifies the best compromises between the $POD$ and the $FAR$.

Last, to be able to analyse the variability of results between reaches (we have a total of 90 reaches affected by the inundation), we also computed the $CSI$ and $BIAS$ reach by reach, and produced two cumulative distribution plots showing these results (Fig. 11). We found that:

- The fit criteria are very sensitive to the $K_{fp}$ value and much less to the $K_{ch}$ value (Fig. 9) : this should not be a surprise given that we deal with the maximum flood extents for calibration, where $K_{ch}$ only plays a minor role. Remember also that (i) we are modelling a very extreme event ($T > 500$ years) with substantial overflowing, and (ii) that we are working with a channel geometry derived from hydraulic geometry relationships. All this contributes to make the estimation of the channel roughness coefficient more difficult.

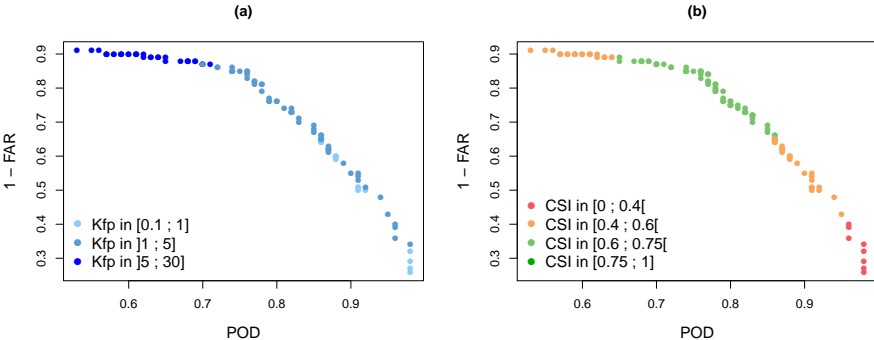

**Figure 10.** Pareto diagram for two forecast scores, $POD$ and $FAR$. $1 - FAR$ is used so that each criterion evolves in the same way, (a) distribution of $K_{fp}$ values and (b) distribution of $CSI$ values according to $POD$ and $1 - FAR$.

- – The $CSI$ clearly shows an optimal zone around $K_{fp} = 5$ and $K_{ch} \leq 10\,\mathrm{m}^{1/3} \cdot \mathrm{s}^{-1}$. The best $CSI$ values (greater than 0.66) correspond to combinations where $0.1 \leq K_{ch} \leq 10$ with $K_{fp} = 5$ or $1 \leq K_{ch} \leq 10$ with $K_{fp} = 4$. Given the equifinality, a good way to choose a combination in this range could be to use the most physical one, which, in this case, would be $K_{ch} = 10$ and $K_{fp} = 5\,\mathrm{m}^{1/3} \cdot \mathrm{s}^{-1}$. Indeed, over the catchment, floodplains mainly consist in $44\%$ non-irrigated arable land, $17\%$ broad-leaved forest and $10\%$ pastures with corresponding roughness coefficients reported in the literature of 8, 2 and $4\,\mathrm{m}^{1/3} \cdot \mathrm{s}^{-1}$, respectively (Grimaldi et al., 2010).

- – Another way to confirm the validity of this choice ($K_{ch} = 10$ and $K_{fp} = 5\,\mathrm{m}^{1/3} \cdot \mathrm{s}^{-1}$) is to look at how this parametrisation behaves at the reach scale. Given a total of 90 reaches, we can compute the $CSI$ and $BIAS$ criteria for each of them and draw a distribution (Fig. 11): we observe that the "optimal" distribution is unbiased and that it represents a solution among the best available for each percentile, we can thus trust this parametrisation as a relatively "all-terrain" one for the Loing catchment.

Last, Fig. 10 provides a good illustration on how parameter sets interact with the $FAR$, $POD$ and $CSI$ criteria: choosing from the parameter sets having the best $CSI$ makes it possible to find a compromise between a high $POD$ and a low $FAR$.

### 4.3 Model behaviour

Figures 12 to 14 provide a further illustration with a colour-coded classification of each reach depending on its $CSI$ and $BIAS$ value. Eleven regions are highlighted and numbered because of their poor performance. The reasons of why MHYST was not able to reproduce the inundation extent in these regions are explained below.

1. For the downstream-most part of the Loing (Fig. 12), the reaches are red or orange because this area is only partially covered by the observation, which stops just after the confluence with the small tributary.

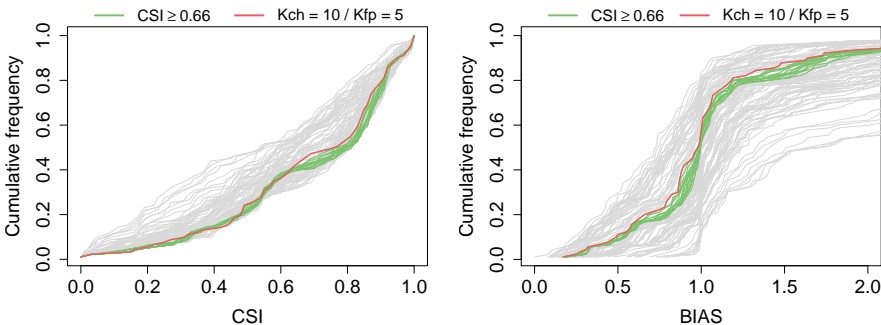

**Figure 11.** Cumulative frequency of $CSI$ and $BIAS$ values for all combinations of parameters and for the 90 affected reaches. Green lines correspond to the best combinations identified in Fig. 9 while the red line refers to the physical parametrisation. The other parameters are displayed in grey.

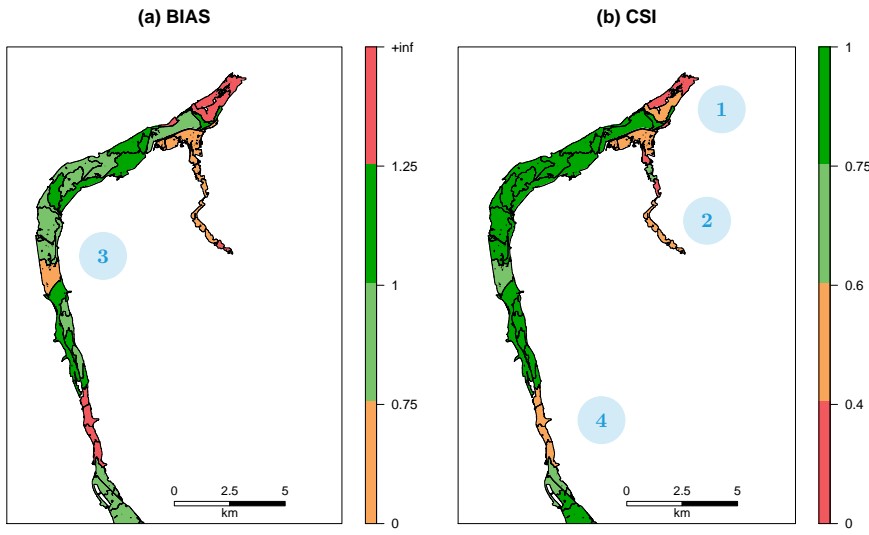

**Figure 12.** Reach-scale performance (a) $BIAS$ and (b) $CSI$ for the physical combination of parameters : $Kch = 10$ and $Kfp = 5m^{1/3} \cdot s^{-1}$, for the downstream part of the catchment. Criteria values have been categorised as follows: excellent (dark green), good (green), average (orange) and poor (red). The black lines delineate the reaches. Locations 1 to 4 correspond to areas where the model struggles to reproduce the observation (orange and/or red zones).

2. The small tributary (Fig. 12) is mainly red or orange for various reasons : downstream, at the confluence, the DEM is full of small high elevation zones (not corrected in the DEM) which the model cannot reach, thus degrading the simulation. Along the tributary, the reason can be either the observed discharge values which seem small compared to the rest of the

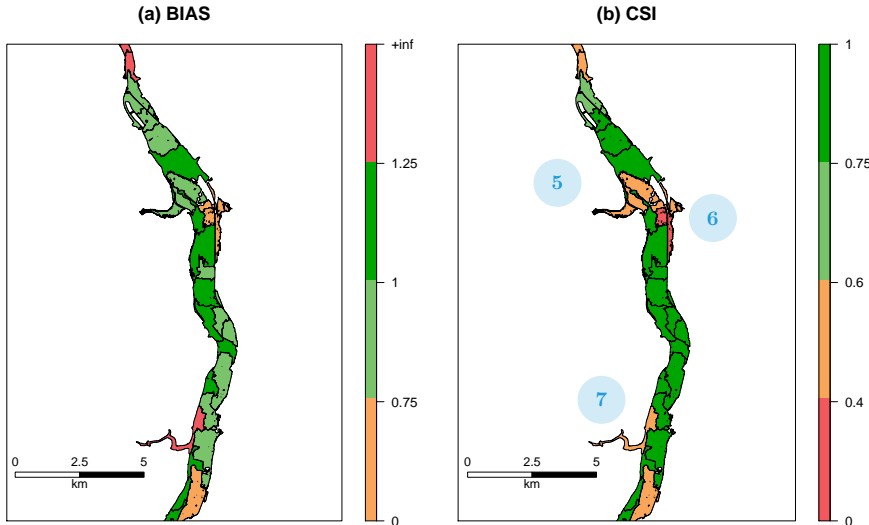

**Figure 13.** Reach-scale performance (a) $BIAS$ and (b) $CSI$ for the physical combination of parameters : $Kch = 10$ and $Kfp = 5m^{1/3} \cdot s^{-1}$, for the center part of the catchment. Criteria values have been categorised as follows: excellent (dark green), good (green), average (orange) and poor (red). The black lines delineate the reaches. Locations 5 to 7 correspond to areas where the model struggles to reproduce the observation (orange and/or red zones).

3. The orange part in the middle of the $BIAS$ map (Fig. 12) is due to the railway tracks which act like a wall in the DEM, preventing the model from reaching the other side (from east to west), where a small tributary, which looks like a partly subterranean urban stream, overflowed in its open air part.

4. Finally, the red and orange zones in the south of the presented map (Fig. 12) correspond to a part of the river where the Loing man-made waterway plays a major role, running parrallel with the main river. This configuration is difficult for MHYST because we only consider the main river, defined by the DEM, with an effective reach-scale geometry and we cannot take into account such specificities, which would require a 2D hydraulic model.

5. The area identified (Fig. 13) shows a slight under-estimation leading to a moderate $CSI$. This issue can be explained by a motorway which is represented in the DEM by a more elevated area. This motorway seperates the reach into two parts linked by artificial openings made by the producers of the DEM. This, as well as the Loing waterway and another road

catchment or simply the effective geometry defined by the model which does not correspond to the actual one. Finally, the upstream part of the tributary is not covered by the observation, which stops in the middle of what MHYST simulated. However, the study zone defined by the Copernicus Emergency Management Service goes further, so we cannot know whether it was not flooded or if the service did not map this part because it was too insignificant.

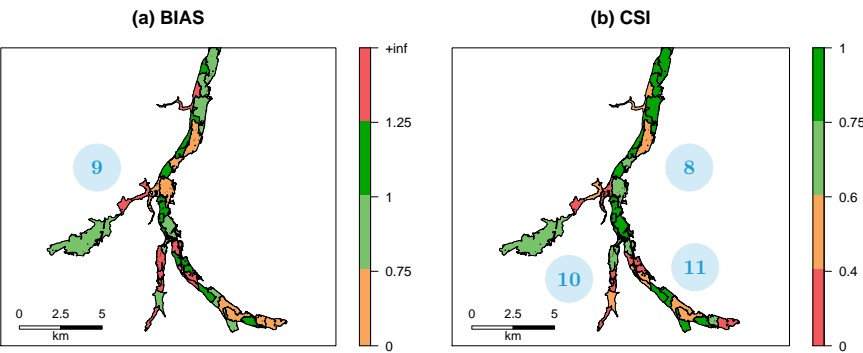

**Figure 14.** Reach-scale performance (a) $BIAS$ and (b) $CSI$ for the physical combination of parameters : $Kch = 10$ and $Kfp = 5m^{1/3} \cdot s^{-1}$, for the upstream part of the catchment. Criteria values have been categorised as follows: excellent (dark green), good (green), average (orange) and poor (red). The black lines delineate the reaches. Locations 8 to 11 correspond to areas where the model struggles to reproduce the observation (orange and/or red zones).

act as dikes that prevent the model from reaching a further part of the reach. The parameterisation of the model is not suitable to address this difficulty.

6. Similarly to the previous area (Fig. 13), a railway crosses the DEM from North to South with only one opening for the water. Given the parameterisation of the model, it is not possible to go over the railway to flood the missed area.

7. In that case (Fig. 13), the model clearly overestimates the flood. The water fills a depression which looks like a tributary but is only a thalweg. Once more, the parameterisation of the model does not provide a adequate representation of this reach.

8. In this area (Fig. 14), MHYST underestimates the inundation extent due to a road that works like a dike. However, with another parameterisation, the model would be able to provide enough water to go over the road.

9. In the western part of the upstream area (Fig. 14), MHYST overestimates the flood because it is a relatively flat zone. The exceeding water, still due to the parameterisation, is thus spread over the area.

10. This area (Fig. 14) is special because the overestimation of MHYST is due to a non-continuous observation map, creating large parts of reaches that are observed dry. However, since MHYST works at the reach scale, it necessarily floods the whole river reach. Moreover, one tributary, the Solin, is not defined in the hydrographic network used by the model, because no observed dicharges were available, whereas it appears in the observed map, leading to an underestimation of the flooded area.

11. The most upstream part of the simulated area (Fig. 14) suffers from an excess of water and a non-continuous observation, leading to similar effects. Moreover, several elevated roads appear in the DEM and force the model to flood the area using artificial openings accross the roads.

In order to complete our interpretation of MHYST behaviour, we conducted two sensitiviy analyses, one with the Morris method (Morris, 1991) and the other with the Sobol method (Sobol, 2001, details can be found in Appendix A). We chose to assess the effect of six potential parameters : $K_{ch}$, $K_{fp}$, $\alpha$, $\beta$, $\delta$ and $\omega$ that may play a major role in the computation of $H_T - Q$ relationships. In both analyses, we found that $\omega$, which parameterises the regionalisation of bankful heights, has the most substantial effect on the performance and that, suprisingly, $K_{fp}$ has no influence at all. As a matter of fact, when we conducted the Sobol analysis with fixed hydraulic geometry parameters, we showed that $K_{fp}$ is consireably more influential than $K_{ch}$. We concluded that the previous results were due to the fact that these sensitivity analyses explore widely the parameters space, and even with reasonable boundaries, they can reach values that may not be consistent with the characteristics of the catchment studied.

## 4.4 Influence of the DEM resolution

It is possible to assess the sensitivity to the DEM with two ways : first by aggregating our DEM from 5 m to various resolutions (10, 25, 50 and 100 m) and then by changing the source of the DEM. Figure 15 provides the $CSI$ scores obtained by the model while changing the resolution. It shows that the resolution has relatively little effect on the optimal value, which varies between 0.65 and 0.69. However, the position of this optimal, i.e. the combination of parameters ($K_{ch}$ and $K_{fp}$) leading to it, changes. We can also see that for some resolutions, such as 25 or 50 m, the equifinality zone is much smaller than the one for the 100-m resolution, for example. If we also look at the "physical" set of parameters we previously identified ($Kch = 10$ and $Kfp = 5m^{1/3} \cdot s^{-1}$), we can see that the $CSI$ reached by the model for this combination varies between the resolutions. Nevertheless, the result still seems satisfying so it could be used as a "default" parameterisation, for instance for ungauged catchments. But this should be tested on other catchments with observed data to lead to a more comprehensive conclusion.

Before using the RGE 5-m DEM from IGN, we tried to use the 25-m EU-DEM from the European Environment Agency, and it showed poorer results, because it was not precise enough. Figure 16 shows the evolution of $CSI$ for the same combinations of parameters as before. We see that the best combinations of parameters only lead to a 0.53 maximal $CSI$, which is more than 10 points below what we can obtain with the RGE DEM. There is also strictly no connection between the best values of $BIAS$ and those of $CSI$, the latter being obtained for a clear overestimation of the flood extent ($BIAS \sim 1.5$). These results are due to the lack of precision of the EU-DEM, which does not distinguish the channel from the floodplain, leading to a 2-km wide channel in some parts of the river.

## 5 Conclusions and outlooks

The objective of this paper was to present and validate a simple hydraulic model for rapid inundation mapping in data-scarce areas. MHYST is based on DEM analyses and simple hydraulic equations, creating a reach-scale relation between the average

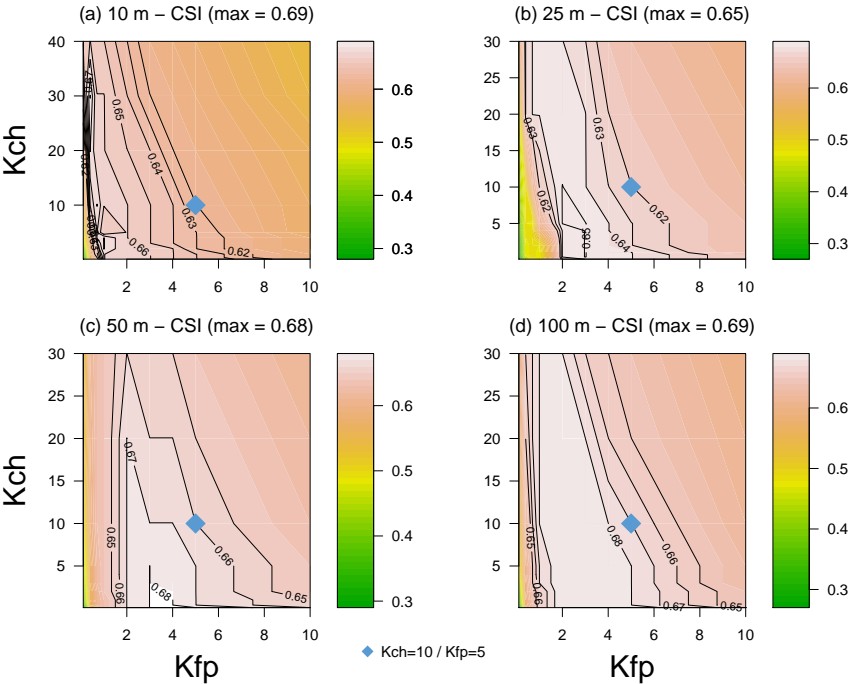

**Figure 15.** $CSI$ scores obtained by the model on the River Loing versus Copernicus data for all the parameter values tested and for various resolutions of the DEM, aggregated from the 5-m resolution DEM : (a) 10 m, (b) 25 m, (c) 50 m and (d) 100 m.

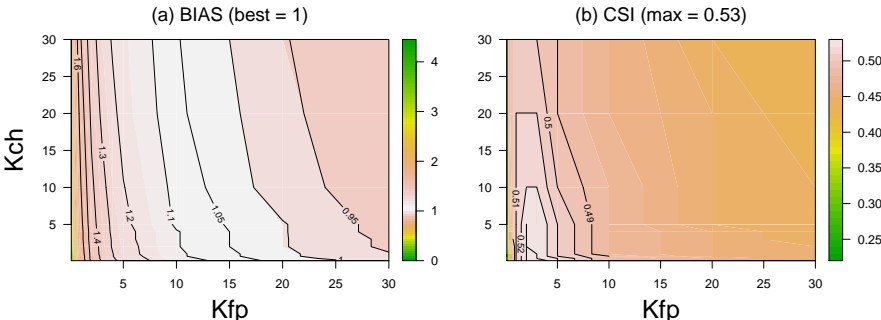

**Figure 16.** (a) $BIAS$ and (b) $CSI$ scores obtained by the model on the River Loing versus Copernicus data for all the parameter values tested and for another source of data : EU-DEM.

discharge and the average "Height Above Nearest Drainage" which can then be used to simulate any event, past or future, as long as streamflow information (observed or simulated) are available. This model was calibrated against an observed exceptional flood which occurred in 2016 on the Loing River near Paris and showed results that are certainly not perfect, but from our point of view and for our objectives quite encouraging. Furthermore, we compared our methodology with the traditional

HAND approach, using a single threshold height of 4m (measured height at the outlet) for the whole catchment. The simple HAND model reached $CSI = 0.49$, $BIAS = 1.55$, $POD = 0.84$ and $FAR = 0.46$. It is clearly penalised by the overestimation (almost $50\%$ of false alarms), which is not surprising according to other studies (Nobre et al., 2016).

The simple structure of MHYST allows it to be used almost anywhere with few data and only two parameters. The model can, however, be used in first approximation, when a lack of time and data restrains the use of a more complex method.

For the sake of honesty, we would like to specify the theoretical limits of the MHYST approach:

– The model equations were solved making the hypothesis of a reach-scale steady uniform flow (probably one of the most simplifying assumptions one can make). This simplification is probably too extreme for highly complex situations, especially in the presence of dikes and bridges. Indeed, on the one hand, the DEM resolution is too coarse to precisely take into account hydraulic structures, and on the other hand, the DEBORD formulation is not sufficient to describe the interaction between the flow and these structures.

– The DEM is a critical part of the model, because geometrical relationships and variables are directly related to the shape and distribution of elevations. Another DEM was actually tested as model input and showed much poorer results.

– Moreover, since the channel geometry was unknown, hydraulic geometry equations were used to assess bankfull height and width, with fixed parameters from another study in the case of height, which may not be the optimum for this catchment, adding its share of uncertainty;

– Finally, there is at this point no continuity equation between reaches, since the calculations were made for each reach separately. Uncertainties may therefore be higher in areas around connection points between reaches, especially if it is a confluence of rivers. One way to adress this issue could be to add a continuity equation between the reaches, which might increase the overall coherence of the flood. However, at this point of the development of the model, we do not have included this specificity.

Thus, the maps produced by MHYST should be seen as a maximum extent of the flood which can be used as a first and rapid estimation. To further test this approach, we consider that attention should first be given to: assessing the impact of the DEM choice, resolution and quality; testing the approach on a range of (less extreme) events and catchments, to better assess the range and stability of its parameters and performance and improving the treatment of possible discontinuities between reaches.

*Data availability.* The IGN DEM cannot be freely downloaded.

Copernicus Emergency Management Service data and the corresponding report can be downloaded at:

http://emergency.copernicus.eu/mapping/list-of-components/EMSN028

French observed discharges can be downloaded at:

http://hydro.eaufrance.fr/indexd.php.

## Appendix A: Sensitivity analysis

In order to assess the sensitivity of the model to its main parameters ($K_{fp}$, $K_{ch}$, $\alpha$, $\beta$, $\omega$, $\delta$), we conducted two sensitivity analyses, using different but complementary well-known methods : Morris (Morris, 1991) and Sobol (Sobol, 2001).

### A1 Morris method

The Morris method (Morris, 1991), which provides a qualification of the effect a parameter can have on the outputs. It is a OAT (one-at-a-time) methodology, which means that the effect of a parameter is measured by changing its value by adding $\pm\Delta$ without modifying the other parameters and by comparing the outputs. In order to provide a relevant analysis, we generated 160 sets of parameters, using the latin hypercube sampling method, which acts as starting points from where the Morris method can assess the significance of parameters by changing their values one-at-a-time. Thus, more than a thousand simulations are

needed to conduct the analysis. By using the 5-m resolution DEM we used in this paper, this study would take several days, if not weeks, to complete. But since we showed that the performance of MHYST did not really change with the resolution, we chose to use a coarser version of our DEM, which was aggregated at a 50-m resolution, by simply averaging the elevations, allowing us to complete this sensitivity analysis in only a few hours. For each permutation and for each parameters, $D_i$, the difference of $CSI$ divided by the computing step, is calculated. The results in terms of means and standard deviations are

presented in Fig. A1. The analysis shows that the model is very sensitive to changes of $\omega$, the exponent in the calculation of the regionalised bankfull width ($W_b$). The most surprising part of the analysis is the fact that $K_{fp}$ has little or no effect on the model, while $K_{ch}$ has a moderate effect. This is contradicted by Fig. 9, which clearly shows that for a given value of $K_{fp}$, the $CSI$ value varies only slightly for a $K_{ch}$ between 0.1 and 20. $K_{fp}$ is, contrary to what the Morris analysis shows, a significant parameter of the model, particularly in a major overflowing event such as the one studied here, where the channel

only represents a fraction of the water.

The problem might be that despite the use of a latin hypercube sampling method, the "good" values of the parameters never meet, *i.e.* when $\omega$ has a sensible value, $K_{fp}$ has not and inversely. And of course, if the $\omega$ value does not coherently represent the channel, the model is not able to conduct a correct simulation (*i.e.* little or no flooding), leading to little or no influence of the $K_{fp}$ parameter.

Moreover, the issue with sensitivity analyses such as the Morris method is that the results can be very different depending on the catchment or the event modelled. Indeed, if the water is concentrated in the channel part for a very steep catchment, a very flat one will on the contrary rely on the floodplains, and so the parameterisation of the model will add more value to $K_{ch}$ or $K_{fp}$. Thus, the conclusions one can make by interpretating one analysis of an example do not necessarily reflect the global behaviour of the model.

### A2 Sobol Method

The Sobol method (Sobol, 2001) is a variance-based sensitivity analysis which aims to compute the fraction of the variance that can be attributed to each parameter. For this study, $2 \times 500$ sets of parameters were randomly chosen with a Latin hypercube

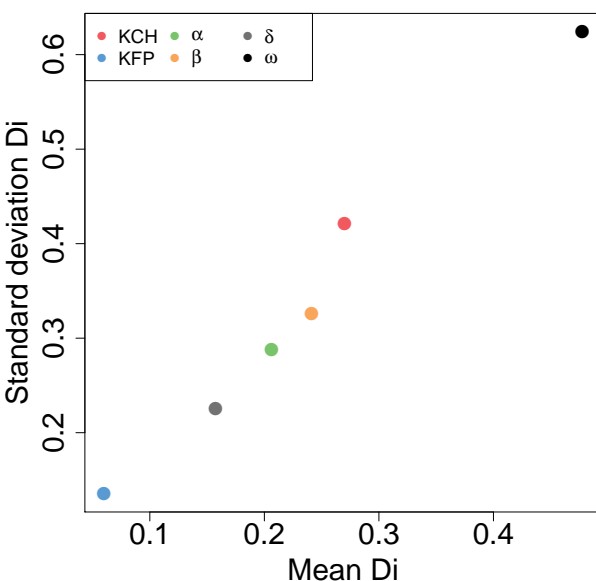

**Figure A1.** Results of the Morris method applied to MHYST with a 50-m resolution DEM on the Loing catchment for the six parameters $(K_{fp}, K_{ch}, \alpha, \beta, \omega, \delta)$.

sampling method, thus creating two $500 \times 6$ matrices, $X_A$ and $X_B$. Each column of $X_A$ has sequentially been substituted by a column of $X_B$, corresponding to one of the six parameters, leading to 6 other matrices. In order to limit the computation time, the interaction of several parameters (*i.e.* subsituting two or more columns of $X_A$ by those of $X_B$) has not been assessed. Indeed, MHYST has been launched with the 4000 sets of parameters, with a resolution of 50 m, which takes longer than the

Morris method that only needed about a thousand simulations. The first-order Sobol indices $S_i$, which indicate the contribution of one parameter to the total variance, and the total-effect indices $S_{-i}$ which calculate the total contribution of one parameter to the variance, including the possible interactions between parameters, have been computed. Then, with a bootstrap re-sampling method, the distributions of $S_i$ and $S_{-i}$ have been assessed, allowing to compute several characteristics such as the bias, the standard deviation and the confidence intervals.

The results of this analysis are presented in Table A1 for $S_i$ and Table A2 for $S_{-i}$. The first-order indices confirm parts of what was concluded from the Morris analysis, interpreting $\omega$ as the most influential parameter, $K_{ch}$ and $\alpha$ as moderately influential and $K_{fp}$ as not influential, despite the observations we made in the article when we calibrated the parameters. The total-effect indices complete the analysis and confirm the conclusions we made with the Morris method, adding $\beta$ to the list of influential parameters.

The distributions of $S_i$ and $S_{-i}$ show that the values calculated are not biased, but the $95\%$ confidence interval is rather large, which means that in some cases, the interpretation may differ. This might explain why when we set values for all downstream hydraulic geometry equations parameters $(\alpha, \beta, \delta, \omega)$ from regionalised studies or observations, $K_{fp}$ has a greater

**Table A1.** Sobol first-order indices for the six parameters of MHYST.

| Parameter | $S_i$ value | bias | std. error | min conf. int. | max conf. int. |
|-----------|-------------|------|-----------|----------------|----------------|
| $K_{ch}$ | 0.121 | 0.004 | 0.193 | -0.149 | 0.392 |
| $K_{fp}$ | 0.043 | -0.004 | 0.065 | -0.071 | 0.156 |
| $\alpha$ | 0.158 | 0.013 | 0.205 | -0.200 | 0.517 |
| $\beta$ | 0.077 | 0.013 | 0.166 | -0.187 | 0.341 |
| $\delta$ | 0.015 | -0.0001 | 0.082 | -0.116 | 0.146 |
| $\omega$ | 0.417 | 0.044 | 0.238 | 0.009 | 0.825 |

**Table A2.** Sobol total-effect index for the six parameters of MHYST.

| Parameter | $S_{-i}$ value | bias | std. error | min conf. int. | max conf. int. |
|-----------|----------------|------|-----------|----------------|----------------|
| $K_{ch}$ | 0.201 | 0.013 | 0.135 | -0.007 | 0.410 |
| $K_{fp}$ | 0.009 | -0.00007 | 0.085 | -0.139 | 0.157 |
| $\alpha$ | 0.238 | -0.002 | 0.156 | -0.038 | 0.514 |
| $\beta$ | 0.167 | 0.001 | 0.128 | -0.054 | 0.389 |
| $\delta$ | 0.047 | -0.001 | 0.068 | -0.060 | 0.156 |
| $\omega$ | 0.476 | -0.003 | 0.22 | 0.120 | 0.832 |

influence which is not highlighted by the sensitivity analyses. These methodologies (Morris, Sobol) indeed explore widely the paremeters space, and even with reasonable boudaries, they can reach values that may not be consistent with the characteristics of the catchment studied. Another limitation is the fact that these analyses are only valid for this particular example (the Loing catchment and the event of May-June 2016). They should ideally be used with a larger set of catchments and events to be

5 reliably trusted.

In order to understand why Morris and Sobol give, contrary to our initial expectation, so little importance to $K_{fp}$, we conducted a quick Sobol analysis with fixed hydraulic geometry parameters, *i.e.* we considered the $\alpha$, $\beta$, $\delta$ and $\omega$ values used in the original study and only made $K_{ch}$ and $K_{fp}$ vary. This time, the results confirm what we observed : $S_{K_{ch}} = 0.15$ and $S_{K_{fp}} = 0.85$, which means that $K_{fp}$ is a major parameter in our situation, and that $K_{ch}$ has a smaller role.

10 The hydraulic geometry parameters are clearly important, but if they are fixed to legitimate values estimated by observations or tables of regionalised values, their impact becomes minor in front of the Strickler coefficients.

*Author contributions.* The model presented in this paper was developed and analysed by Cédric Rebolho during his PhD work. He also wrote the manuscript which was corrected by Vazken Andréassian and Nicolas Le Moine.

*Competing interests.* The authors declare that they have no conflict of interest.

*Acknowledgements.* The first author was funded by a grant from the AXA Research Fund. Thanks are extended to Rafal Zielinksi, who helped us access data from the Copernicus Emergency Management Service. We would also like to thank the AXA Global P&C research team for their advice and our discussions on the development of simple conceptual inundation models. The MHYST model was developed using R (R Core Team, 2015) and GFortran, Gnu compiler collection (gcc) Version 4.9.2.

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
