# Peer review of "Inundation mapping based on reach-scale effective geometry"

_Hydrology and Earth System Sciences, 2018_

## Referee Comment (RC1) · R. Romanowicz (Referee) · 2 May 2018

The paper presents a new approach to the derivation of inundation maps in data-scarce areas, the so - called MHYST approach. The approach applies concepts of hydraulic geometry to describe channel geometry and DEM-derived elevation geometry on the floodplains at a reach-scale.

It is an engineering approach to inundation mapping and is a good counter-part to "model-based" mapping which sometimes relies more on modelling results than on reality. The approach is based on an experimental study proposing a parameterisation of a turbulent momentum exchange between the channel and the floodplain. The uniform flow assumptions allow the streamflow value to be defined from the conveyance

capacity and the channel slope.

The operation of the approach on a reach scale sums up to the derivation of a discharge- water level threshold relationship, assuming it is one-to-one. The approach's main advantage is its simplicity and speed of computations and it can be used as the first approximation of inundation mapping.

The MHYST approach is tested on one extreme event that occurred in May-June 2016 in France. The River Loing, a tributary of the River Seine, was used as a case study. The inundation data were obtained from observations, thus omitting the application of a hydrological model.

The validation procedure is based on the maximum inundation area. However, it is not clear which data were chosen for the calibration and validation stages. The authors are asked to explain this point in detail. A number of different criteria were used but the description is very vague. The choice of roughness coefficients for a channel and a floodplain raises the most concern.

The authors present the drawbacks of the approach proposed, which covers most of my concerns.

The illustration map presented in Fig. 11 does not give enough detail. Perhaps the authors could present a larger scale map focussing on some specific area?

The authors stress the importance of the DEM in the derivation of inundation maps. Perhaps some sensitivity studies could be performed to assess that influence in a quantitative way.

I am also worried about the possible inconsistency between flood inundation assessment on adjacent reaches of the same river. The authors are asked to discuss that point. Finally, the authors are asked to compare the inundation mapping using MHYST with the straight-forward DEM based mapping in order to show the advantages of the proposed method to the simplest possible, "filling the volume" approach.

---

## Author Comment (AC1) · 16 May 2018

**Answer to the review comments by Renata Romanowicz**

The authors would like to thank Prof. Renata Romanowicz for her precise comments on the article which will undoubtedly help to improve the quality of the manuscript. We hope this discussion will answer the concerns about the methodology.

[Figure]

Specific comments

1. Page 10, section 4: The validation procedure is based on the maximum inundation area. However, it is not clear which data were chosen for the calibration and validation stages. The authors are asked to explain this point in detail.

    The description of the validation procedure was indeed slightly ambiguous. The title of the subsections should be changed to "Calibration procedure" and "Parameterisation" since no real validation was performed in the study, because of a lack of data. We only had one inundation event on the Loing catchment which allowed us to parameterise the model but not to validate it on a second event. The data used for the calibration were the observed discharges that led to the simulated inundation map which was then compared to the observed inundation extent from the activation EMSN028 of the Copernicus Emergency Management Service.

    That being said, we parameterise our model at the catchment scale (i.e. with one set of parameters for the whole catchment), which is different from using the best combination calculated for each reach (which would be a true calibration). Thus, when we analyse the reach-scale performance with the catchment-scale parameters, we do some sort of validation analysis, proving that with one combination (which is not the reach-scale optimum) we can still perform well.

2. Pages 10, 11 and 12, section 4.1: A number of different criteria were used but the description is very vague.

    To evaluate the performance of our methodology, we based our analysis on four classic criteria extracted from inundation and forecast studies. The $POD$ (Probability of detection), which is also called $Correct$ (Alfieri et al., 2014) or $M1$ (Teng et al., 2015), calculates the percentage of observed inundated pixels intersected by the simulation map. Its main drawback is that it does not take into account the

false alarms and thus it can give good results for a clearly overstimating inundation extent. On the contrary, the $FAR$ (False alarm ratio) or $M2$ (Teng et al., 2015) computes the proportion of cells wrongly flooded by the model. But similarly, if the model does not flood anything, the $FAR$ can reach its optimal value. The $CSI$, also known as $Fit$, $F$ index or $FAI$ (Alfieri et al., 2014; Bates and De Roo, 2000; Falter et al., 2015), is a criterion which tries to give an overall performance of the simulation by calculating the percentage of correctly flooded cells above the total number of flooded cells (observed and simulated). In this way, the score is penalised by the over- and under-estimation. However, this criterion does not specify if the model is over- or under-estimating the observed extent. This is why we also looked at the $BIAS$, which computes the ratio between the simulated and observed flooded cells. If it is above $1$, the model over-estimates, and if it is below $1$, it under-estimates. However, a value of $1$ does not equal a perfect simulation since there may be a balance between the misses and the false alarms. This complete description of the criteria shall be added in the manuscript.

3. Page 15, section 4.2 : The illustration map presented in Fig. 11 does not give enough detail. Perhaps the authors could present a larger scale map focussing on some specific area?

We could, for example, present the overall view of the flood and then focus on the downstream part of the catchment (Fig.1), which is quite interesting. Indeed, there is a majority of green reaches but also some examples of why the model can perform badly.

- For the most downstream part of the Loing, the reaches are red or orange because this area is only partially covered by the observation, which stops just after the confluence with the small tributary.
- The small tributary is mainly red or orange for various reasons : downstream, at the confluence, the DEM is full of small high elevation zones (not

corrected in the DEM) which the model cannot reach, thus degrading the simulation. Along the tributary, the reason can be either the observed discharge values which seem small compared to the rest of the catchment or simply the effective geometry defined by the model which do not correspond to the actual one. Finally, the upstream part of the tributary is not covered by the observation, which stops in the middle of what MHYST simulated. However, the study zone defined by the Copernicus Emergency Management Service goes further, so we cannot know if it was not flooded or if the service did not map this part because it was too insignificant.

- The orange part in the middle of the $BIAS$ map is due to a railway which acts like a wall in the DEM, preventing the model from reaching the other side (from east to west), where a small tributary, which looks like a partly subterranean urban stream, overflowed in its open air part.

- Finally, the red and orange zones in the south of the presented map correspond to a part of the river where the Loing man-made waterway plays a major role, and is parrallel to the main river. This configuration is difficult for MHYST because we only consider the main river, defined by the DEM, with an effective reach-scale geometry and we cannot take into account such specificities, like a 2D hydraulic model would.

These comments should be added in the article with the corresponding map.

4. Page 14, Section 5 : The authors stress the importance of the DEM in the derivation of inundation maps. Perhaps some sensitivity studies could be performed to assess that influence in a quantitative way.

We can assess the sensitivity to the DEM with two ways : first by aggregating our DEM from $5$ m to various resolutions (10, 25, 50 and 100 m) and then by changing the source of the DEM. Figure 2 and 3 provides the $CSI$ scores obtained by the model while changing the resolution. It shows that the resolution has relatively

little effect on the optimal value, which varies between 0.65 and 0.69. However, the position of this optimal, i.e. the combination of parameters ($K_{ch}$ and $K_{fp}$) leading to it, changes. We can also see that for some resolutions, such as 25 or 50 m, the equifinality zone is much smaller than the one for the 100-m resolution, for example. If we also look at the "physical" set of parameters we identified in the article ($Kch = 10$ and $Kfp = 5m^{1/3} \cdot s^{-1}$), we can see that the $CSI$ reached by the model for this combination varies between the resolutions. Nevertheless, the result still seems satisfying so it could be used as a "default" parameterisation, for instance for ungauged catchments. But this should be tested on other catchments with observed data to lead to a more comprehensive conclusion.

Before using the RGE 5-m DEM from IGN, we tried to use the 25-m EU-DEM from the European Environment Agency, and it showed poorer results, because it was not precise enough. Figure 4 shows the evolution of $CSI$ for the same combinations of parameters as before. We see that the best combinations of parameters only lead to a 0.53 maximal $CSI$, which is more than 10 points below what we can obtain with the RGE DEM. There is also strictly no connection between the best values of $BIAS$ and those of $CSI$, the latter being obtained for a clear overestimation of the flood extent ($BIAS \sim 1.5$). These results are due to the lack of precision of the EU-DEM, which does not distinguish the channel from the floodplain, leading to a 2-km wide channel in some parts of the river.

5. Page 14, Section 5 : I am also worried about the possible inconsistency between flood inundation assessment on adjacent reaches of the same river. The authors are asked to discuss that point.

As the overall view of the simulated inundation extent shows in the article, two adjacent reaches may not have the same $CSI$, one being very good while the other has a poor performance, despite their proximity. This is mainly due to local specificities, such as the railway in the DEM, the man-made waterway which is not taken into account in the model, or an urban area which is harder to model

because it is not natural and can have been modified by man.

One way to adress this issue could be to add a continuity equation between the reaches, which might increase the overall coherence of the flood. However, at this point of the development of the model, we do not have included this specificity.

6. Finally, the authors are asked to compare the inundation mapping using MHYST with the straight-forward DEM based mapping in order to show the advantages of the proposed method to the simplest possible, "filling the volume" approach.

We compared our methodology with the traditional HAND approach, using a single threshold height for the whole catchment. According to the observed data, the maximal height reached at the outlet is 4 m, so we used this threshold to simulate an inundation map. The simple HAND model reached $CSI = 0.49$, $BIAS = 1.55$, $POD = 0.84$ and $FAR = 0.46$. The HAND model is clearly penalised by the overestimation (almost $50\%$ of false alarms), which is not surprising according to other studies (e.g. Nobre et al., 2016).

In comparison, for the "physical" set of parameters identified with MHYST, ($K_{ch} =$ 10 and $K_{fp} = 5$), we obtained $CSI = 0.66$, $BIAS = 0.86$, $POD = 0.74$ and $FAR = 0.14$. Here, the issue is the underestimation, but MHYST reduces the false alarm ratio while increasing the overall performance ($CSI$), which means that the part of missed observed flooded cells decrease only slightly (we reduce $30\%$ of $FAR$ while losing only $10\%$ of POD).
* * *
**BIAS**

**CSI**

**Fig. 1.** Reach-scale performance for the physical combination of parameters, for the downstream part of the catchment.

[Figure]

**Fig. 2.** CSI scores obtained by the model on the River Loing versus Copernicus data for all the parameter values tested and for a 5-m resolution DEM.

**10 m – CSI (max = 0.69)**

**25 m – CSI (max = 0.65)**

**50 m – CSI (max = 0.68)**

**100 m – CSI (max = 0.69)**

◆ Kch=10 / Kfp=5

Kch

Kfp

**Fig. 3.** CSI scores obtained by the model on the River Loing versus Copernicus data for all the parameter values tested and for various resolution of the DEM.

[Figure]

**Fig. 4.** CSI scores obtained by the model on the River Loing versus Copernicus data for all the parameter values tested and for another source of data : EU-DEM.

---

## Referee Comment (RC2) · Anonymous Referee #2 · 14 Jul 2018

*Hydrol. Earth Syst. Sci. Discuss.*

1st revision of Manuscript Number:  doi.org/10.5194/hess-2018-146, 2018

Title:  Inundation mapping based on reach-scale effective geometry

**General comments**

This paper introduces the development of a novel conceptual inundation mapping procedure based on the well-known DEM-based HAND concept and standard hydraulic geometry functions coupled with 1-D steady-state flow equations.  The paper, which is well organized and written, deals with a current subject matter that has been covered lately in an increasing number of papers published primarily in *Journal of Hydrology, Natural Hazards, and Journal of the American Water Resources Association*, to name a few.

The procedure known as MHYST is not meant to be a surrogate to physics-based hydrodynamic modelling of flood inundation mapping, but it may turn out to be a valuable alternative to computationally-intensive models and in data-poor regions.  MHYST has the potential to be applied under different reach-scale and flood plain conditions; that is from natural to urban river corridors.  The procedure was validated using an extreme flood event in France. That being mentioned, the procedure has yet to be tested under these aforementioned conditions and, more importantly, to prove it can be useful for a wider range of flooding events.  Furthermore, I would argue there is a need to:

(i)     conduct formal sensitivity and uncertainty analyses (*e.g.*, Morris and Sobol) of key parameters (*e.g.*, $K_{fp}$, $K_{ch}$, $\alpha$, $\beta$, $\omega$, $\delta$) and

(ii)    include in the paper a comparison between an actual and a predicted inundation mapping of several continuous reaches (*i.e.*, flooding extent) for one, two or three days; that is the information required by key stakeholders and elected officials.

At this point, I feel the authors should be given a fair chance to respond to the above comments and those introduced in the next section since I feel the paper has the potential to be a valuable contribution to *Hydrology and Earth System Sciences*. Thus, for the time being, I would say that revisions are necessary and required before the paper could be considered as a good technological contribution to the hydrological community.

**Specific comments**

P.3, L.28  Please specify the algorithm behind the flow direction function available in ESRI ArcGIS? Which software version?

P.8, L19   What is the vertical resolution of the 5-m DEM?

P.15 Given Fig. 11, is there any general observations about why and where MHYST performed either poorly or satisfactorily? Furthermore, the performance accounts for how many days? Is it only the flood mapping associated with the peak flow?

**Editorial suggestions**

P.2, L.5 Please insert « to » between « need » and « have »

P.3, L.5 I suggest adding the following reference: McGrath *et al.* (2018). A comparison of simplified conceptual models for rapid web-based flood inundation mapping. *Natural Hazards* https://doi.org/10.1007/s11069-018-3331-y

P.3, L.21 Please replace « relations » by « relationships » or « functions » and do so throughout the text (e.g., P.3, L.22, P.3, L23, P.4, L.10, P4. L13…so on so forth)

P.8, L.4 The complete citation is: Zheng, X., D.G. Tarboton, D.R. Maidment, Y.Y. Liu, and P. Passalacqua. 2018. "River Channel Geometry and Rating Curve Estimation Using Height above the Nearest Drainage." *Journal of the American Water Resources Association* 1–22. https://doi.org/10.1111/1752-1688.12661.

**Figures**

Fig.3 For completeness sake, please identify $A_b$, $A_{ch}$ and $A_{fp}$, I know it is trivial, but the figure might as well be as explicit as it can be.

Fig. 4 Please hyphenate: « 5-m depressionless DEM »

Fig. 11 The resolution of this map could be improved, perhaps one image per page.

**Tables**

**Answer to traditional questions**

Is the paper free of errors in logic?

- Yes

Do the conclusions follow from the evidence?

- Yes.

Are alternative explanations explored as appropriate?

- Yes.

Are biases, limitations, and assumptions clearly stated, and uncertainty quantified?

- Yes.

Is methodology explained in sufficient detail so that the paper's scientific conclusions could be tested by others?

- Yes

Is previous work and current understanding cited and represented correctly?

- Yes.

Is information conveyed clearly enough to be understood by the typical reader?

- Yes

Are all figures and tables necessary, appropriate, legible, and annotated (as appropriate)?

- Yes.

---

## Referee Comment (RC3) · Anonymous Referee #2 · 18 Jul 2018

I would like to thank the authors for their valiant effort to answer quickly my questions and comments. I did appreciate the approach and results of the application of the Morris method. And, yes, results of a sensitivity analysis will depend on the case study. That being said, they were somewhat silent on why they opted out of applying the Sobol method to further our understanding of MHYST. Why wasn't it undertaken?

---

## Author Comment (AC2) · 18 Jul 2018

**Answer to the review comments of Reviewer#2**

The authors would like to thank Reviewer#2 for his analyses, questions and suggestions which will help to improve the quality of the manuscript, and the further developments of the method. We hope this discussion will answer the concerns about the methodology. Comments from the reviewer are in blue while our answers are in black.

General comments

 The procedure known as MHYST is not meant to be a surrogate to physicsbased hydrodynamic modelling of flood inundation mapping, but it may turn out to be a valuable alternative to computationally-intensive models and in data-poor regions. MHYST has the potential to be applied under different reach-scale and flood plain conditions; that is from natural to urban river corridors. The procedure was validated using an extreme flood event in France. That being mentioned, the procedure has yet to be tested under these aforementioned conditions and, more importantly, to prove it can be useful for a wider range of flooding events.

Reviewer#2 has perfectly underlined the challenges of our methodology. MHYST has a potential that still needs to be validated on a larger number of situations (urban areas, floodplains, mountainous regions, flat areas...) and on different types of events (small overflowing, flash floods...). This article provides the description of the methodology and its application on a major overflowing event on a catchment presenting both urban areas and floodplains. However, due to the scarcity of observed inundation mapping with corresponding distributed observed streamflows, MHYST was only applied on one example. Nonetheless, its application to a forecasting context on a larger sample of catchments and events is part of an ongoing research project and a beginning PhD thesis.

• There is a need to conduct formal sensitivity and uncertainty analyses (*e.g.*, Morris and Sobol) of key parameters (*e.g.*,  $K_{fp}$ ,  $K_{ch}$ ,  $\alpha$ ,  $\beta$ ,  $\omega$ ,  $\delta$ ).

In order to assess the sensitivity of the model to its parameters ( $K_{fp}$ ,  $K_{ch}$ ,  $\alpha$ ,  $\beta$ ,  $\omega$ ,  $\delta$ ), we used the Morris method (Morris, 1991), which provides a qualification of the effect a parameter can have on the outputs. It is a OAT (one-at-atime) methodology, which means that the effect of a parameter is measured by changing its value by adding  $\pm \Delta$  without modifying the other parameters and by comparing the outputs. In order to provide a relevant analysis, we generated 160 sets of parameters, using the latin hypercube sampling method, which act as start points from where the Morris method can assess the significance of parameters
by changing their values one-at-a-time. Thus, more than a thousand simulations are needed to conduct the analysis. By using the 5-m resolution DEM we used in this article, this study would take several days, if not weeks, to complete. Since we showed in the response to Reviewer#1 (Renata Romanowicz) that the performance of MHYST did not really change with the resolution, we chose to use a coarser version of our DEM, which was aggregated at a 50-m resolution, by simply averaging the elevations, allowing us to complete this sensitivity analysis in only a few hours. For each permutation and for each parameters,  $D_i$ , the difference of CSI divided by the computing step, is calculated. The results in terms of means and standard deviations are presented in Fig.1. The analysis shows that the model is very sensitive to changes of  $\omega$ , the exponent in the calculation of the regionalised bankfull width  $(W_b)$ . The assessment of  $W_b$  seems to be a major part of the model, but it is also the easiest, because it is easily feasible to compute a relationship between widths derived from satellite images of the river and drainage areas, which does not need any calibration. The most surprising part of the analysis is the fact that  $K_{fp}$  has little or no effect on the model, while  $K_{ch}$ has a moderate effect. This is contradicted by Fig.8 of the article, which clearly shows that for a given value of  $K_{fp}$  (e.g. 2), the CSI value varies only slightly for a  $K_{ch}$  between 0.1 and 20.  $K_{fp}$  is, contrary to what the Morris analysis shows, a significant parameter of the model, particularly in a major overflowing event such as the one studied in the article, where the channel only represents a fraction of the water.

The problem might be that despite the use of a latin hypercube sampling method, the "good" values of the parameters never meet, *i.e.* when  $\omega$  has a sensible value,  $K_{fp}$  has not and inversely. And of course, if the  $\omega$  value does not coherently represent the channel, the model is not able to conduct a correct simulation (*i.e.* little or no flooding), leading to little or no influence of the  $K_{fp}$  parameter.

Moreover, the issue with sensitivity analyses such as the Morris method is that
the results can be very different depending on the catchment or the event modelled. Indeed, if the water is concentrated in the channel part for a very steep catchment, a very flat one will on the contrary rely on the floodplains, and so the parameterisation of the model will add more value to  $K_{ch}$  or  $K_{fp}$ . Thus, the conclusions one can make by interpretating one analysis of an example do not necessarily reflect the global behaviour of the model.

This analysis and the discussion about its conclusions will be added in the manuscript in order to discuss the sensitivity of the parameters.

• There is a need to include in the paper a comparison between an actual and a predicted inundation mapping of several continuous reaches (*i.e.*, flooding extent) for one, two or three days; that is the information required by key stakeholders and elected officials.

In this paper, as we explained in Section 4.1, P.10, L.8-11, we used an observed maximum inundation extent provided by the Copernicus Emergency Management Service. This map is a composite for several dates and it only represents the inundation associated to the peak flow and is not dated. No other sources of data could provide dated observations so we had no choice but to use these data to calibrate our model, and we did not try to validate the temporal dynamic of the flood, but only its largest area.

Reviewer#2 highlights here a global issue that exists, at least in France: the scarcity of dated observed inundation maps. Generally, observation data are provided for the maximum extent only. In most cases, dated maps corresponding to an event are generated using a complex 2D hydraulic model, which can be really precise, but does not provide observed data. Because our article deals with the development of MHYST we needed to validate it with a comparison to an observed inundation map, not to a simulated one.

A solution to this issue could be the use of insurance damages data, which are

HESSD
relatively precise dated data. We did try to use that kind of information in order to calibrate our model and we found that it leads to different issues : comprehensiveness of the data (the damages of one insurer are not representative of a whole area), different insurance policies (cars, house, flat...), rights of use etc.

Given that there are no day by day observed inundation maps, we will not be able to assess the temporal performance of our model.

Specific comments

• P.3,L.28 Please specify the algorithm behind the flow direction function available in ESRI ArcGIS? Which software version?

We used the D8 method from the Flow Direction function provided by ArcGIS 10.3. It computes the drainage direction by calculating the steepest slope from the eight possible directions for a given cell. This information will be added in the manuscript.

• P.8,L.19 What is the vertical resolution of the 5-m DEM?

We used a 5-m resolution DEM from IGN (the French national institute for geographic information), which has a vertical resolution of 0.01m. This information will be added in the manuscript.

• P.15 Given Fig.11, is there any general observations about why and where MHYST performed either poorly or satisfactorily ? Furthermore, the performance accounts for how many days? Is it only the flood mapping associated with the peak flow ?

Like we said in Section 4.1, P.10, L.8-11, and in the General comments section, the observed inundation mapping represents a "maximum flood extent", *i.e.* the map associated to the peak flow. Since no other sources of data were available,

**HESSD**
we chose to calibrate our model by using this image as a reference, and by combining all our simulated maps (which are produced for each day of the flood) into one maximum simulated extent.

However, we agree that it would be better to assess the performance day by day, but such maps are very scarce, at least in France, where our study was conducted.

Reviewer#1 (Renata Romanowicz) was also concerned by the details of Fig.11, and asked for a larger scale map that would focus on some specific area. We chose to respond to her concerns by focussing on the most downstream part of the flood and by identifying and explaining the reasons why the model had trouble modelling some areas. We finally propose to replace Fig.11 by three figures (Fig.2-4) focussing on three major parts of the catchment (and thus improving the resolution of the maps), and to identify and explain on each map the difficulties MHYST can have. The comments associated to each number in the maps are :

Figure 2:

- 1. For the most downstream part of the Loing, the reaches are red or orange because this area is only partially covered by the observation, which stops just after the confluence with the small tributary.
- 2. The small tributary is mainly red or orange for various reasons : downstream, at the confluence, the DEM is full of small high elevation zones (not corrected in the DEM) which the model cannot reach, thus degrading the simulation. Along the tributary, the reason can be either the observed discharge values which seem small compared to the rest of the catchment or simply the effective geometry defined by the model which do not correspond to the actual one. Finally, the upstream part of the tributary is not covered by the observation, which stops in the middle of what MHYST simulated. However, the study zone defined by the Copernicus Emergency Manage-
ment Service goes further, so we cannot know if it was not flooded or if the service did not map this part because it was too insignificant.

- 3. The orange part in the middle of the *BIAS* map is due to a railway which acts like a wall in the DEM, preventing the model from reaching the other side (from east to west), where a small tributary, which looks like a partly subterranean urban stream, overflowed in its open air part.
- 4. Finally, the red and orange zones in the south of the presented map correspond to a part of the river where the Loing man-made waterway plays a major role, and is parrallel to the main river. This configuration is difficult for MHYST because we only consider the main river, defined by the DEM, with an effective reach-scale geometry and we cannot take into account such specificities, like a 2D hydraulic model would.

Figure 3:

- 5. The area identified shows a slight under-estimation leading to a moderate *CSI*. This issue can be explained by a motorway which is represented in the DEM by a more elevated area. This motorway seperates the reach into two parts linked by artificial openings made by the producers of the DEM. This, as well as the Loing waterway and another road act as dikes that prevent the model from reaching a further part of the reach. The parameterisation of the model is not suitable to address this difficulty.
- 6. Similarly to the previous area, a railway crosses the DEM from North to South with only one opening for the water. Given the parameterisation of the model, it is not possible to go over the railway to flood the missed area.
- 7. In that case, the model clearly overestimates the flood. The water fills a depression which looks like a tributary but is only a thalweg. Once more, the parameterisation of the model does not provide a adequate representation of this reach.

HESSD
Figure 4:

- 8. In this area, MHYST underestimates the inundation extent due to a road that works like a dike. However, with another parameterisation, the model would be able to provide enough water to go over the road.
- 9. In the western part of the upstream area, MHYST overestimates the flood because it is a relatively flat zone. The exceeding water, still due to the parameterisation, is thus spread over the area.
- 10. This area is special because the overestimation of MHYST is due to a noncontinuous observation map, creating large parts of reaches that are observed dry. However, since MHYST works at the reach scale, it necessarily floods the whole river reach. Moreover, one tributary, the Solin, is not defined in the hydrographic network used by the model, because no observed dicharges were available, whereas it appears in the observed map, leading to an underestimation of the flooded area.
- 11. The most upstream part of the simulated area suffers from an excess of water and a non-continuous observation, leading to similar effects. Moreover, several elevated roads appear in the DEM and force the model to flood the area using artificial openings accross the roads.

**Editorial suggestions**

We would like to thank Reviewer#2 for his remarks. We will certainly add the McGrath *et al.* (2018) reference, which is totally relevent. Concerning the Zheng *et al.* (2018) reference, it was still in review when we submitted this article, and we will of course update the citation now that it is finally published.

**Figures**
• Fig.3 For completeness sake, please identify  $A_b$ ,  $A_{ch}$  and  $A_{fp}$ , I know it is trivial, but the figure might as well be as explicit as it can be.

In order to not overload the figure, which is already quite dense, we propose to add Fig.5 into the manuscript : Fig.5 describes a typical cross-section and identifies the variables requested.

• Fig.11 The resolution of this map could be improved, perhaps one image per page.

See the last point of Specific Comments.
Fig. 1. Results of the Morris method applied to MHYST with a 50-m resolution DEM on the Loing catchment for the six parameters.

---

## Author Comment (AC3) · 25 Jul 2018

**Answer to the review comments of Reviewer#2**

[revised manuscript text omitted]

These results will naturally be added in the manuscript to complete our analysis of MHYST's behaviour on the Loing catchment.

146, 2018.

---

## Referee Report (RR1)

*Hydrol. Earth Syst. Sci. Discuss.*

2nd revision of Manuscript Number:  doi.org/10.5194/hess-2018-146, 2018

Title:  Inundation mapping based on reach-scale effective geometry

**General comments**

The authors made a valiant effort to answer the major concerns I had raised in my first review of the paper.  I have gone through all their answers to my specific comments and editorial comments and I am generally satisfied.

In the end, the paper introduces the development of a novel conceptual inundation mapping procedure based on the well-known DEM-based HAND concept and standard hydraulic geometry functions coupled with 1-D steady-state flow equations. The sensitivity and uncertainty analysis represents a nice addition to the paper, providing useful information for future application of the inundation mapping methodology. A major event in France was used to provide a calibration case study.

That being mentioned, the authors should provide a discussion on the fact that a DEM model of a river network does not always provide the geodesic elevation of the river bed. I did not see that discussion in the paper anywhere!

At this point, I recommend minor revision based on the specific comments and editorial suggestions listed below. Once these are addressed by the authors, I believe the paper will be of interest to readers of *Hydrology and Earth Systems Sciences*.

**Specific comments**

P.2, L.7    Please complete the following:  (*e.g.*, ?)

P.18, L4   Please insert bibliographical references after Morris and Sobol.

P.19        The authors state: « …it is also the easiest, because it is easily feasible to compute a relationship between widths derived from satellite images of the river and drainage areas, which does not need any calibration. »

> I disagree. It is difficult to assess the bankfull width of a channel using solely satellite derived data. To my knowledge, any relationship derived using the latter approach requires calibration with *in situ* bathymetric data.  The authors should not oversimplify reality because it suits their argument well! Be frank; that is the least the authors can do.

P.27        Please delete Zheng *et al*. (2017).

Fig. 1    Please in frames (c) and (d), all the cells of the top row (those with a drainage area of 1) should be in pink according to the flow direction matrix displayed in frame (b). Please verify carefully this figure.

Fig. 9    Insert in the caption add to the figure: (a) Bias contour lines and (b) CSI contour lines for various values of $K_{fp}$ and $K_{ch}$.

Fig. 10    The caption is incomplete; (a) and (b) must be defined

Fig. 11    Please modify the figure caption as follows:  Cumulative frequency of CSI and Bias values

Fig. 12    Please define in the figure caption the identified locations 1 through 4. Also, identify each map in the caption as well (*e.g.*:  (a) Bias, (b) CSI). Moreover, each time a location is discussed in the text, remind the reader of the Figure associated with the region. Moreover, each time a location is discussed in the text, remind the reader of the Figure associated with the region.

Fig. 13    Please define in the figure caption the identified locations 5 through 7. Also, identify each map in the caption as well (*e.g.*:  (a) Bias, (b) CSI). Moreover, each time a location is discussed in the text, remind the reader of the Figure associated with the region.

Fig. 14    Please define in the figure caption the identified locations 9 through 11. Also, identify each map in the caption as well (*e.g.*:  (a) Bias, (b) CSI). Moreover, each time a location is discussed in the text, remind the reader of the Figure associated with the region.

Fig. 15    Each contour map should be labeled (a) 10m through (d) 100 m in the figure and defined in the caption.

Fig. 16    Identify each map in the figure and in the caption (*e.g.*:  (a) Bias, (b) CSI).

---

## Author Response (AR2)

**Answer to the review comments by Referee#2**

The authors would like to thank Referee#2 for his/her comments.

**General comments**

We agree with Referee#2 about the fact that DEMs do not correctly represent the river bed. Indeed, the lasers generally used to create DEMs cannot penetrate the water surface more than a few centimeters. This is why we chose to add a subgrid representation of the channel inside MHYST. We simplified the geometry as a rectangle whose height ($h_b$) and width ($W_b$) are calculated from hydraulic geometry equations (see Eq. 3 and 4). While we are aware that this is a large simplification of reality, it is a convenient method which allows to take into account the contribution of the channel in the model.

    **Added/Modified :** The channel geometry is already taken into account inside the model. We added the following sentences to better explain our methodology. *P.5, L. 4 : "Indeed, in situ bathymetric data are quite scarce and red lasers cannot penetrate the water surface more than a few centimeters, which means that the real elevation of the river bed is mostly not correctly represented in the DEMs. This is why we chose to use downstream hydraulic geometry equations to estimate these geometric parameters, assuming a rectangular channel, the size of which depends on the upstream drainage area (Eq. 3 and 4)."*

**Specific comments**

- All figures and errors have been corrected according to Referee#2's comments.

- Concerning the estimation of the bankfull river width, we did use satellite images to compute a hydraulic geometry equation between the width and the upstream drainage areas. While we do not deny that it is clearly a simplification of reality, we assumed it was a better proxy than the use of coefficients calibrated over American rivers in the study of Blackburn-Lynch *et al.* (2017). As for the bankfull height, no local study being available, we did use the coefficients of the latter article.

    **Added/Modified :** *In order to satisfy Referee#2's comments, we decided to remove the contentious sentence, which does not alter the meaning of the analysis.*

[revised manuscript text omitted]